# Catalytic enantioselective reductive alkynylation of amides enables one-pot syntheses of pyrrolidine, piperidine and indolizidine alkaloids

Fang-Fang Xu [1], Jin-Quan Chen [1], Dong-Yang Shao [1] & Pei-Qiang Huang [1] ✉

The primary objective in synthetic organic chemistry is to develop highly efficient, selective, and versatile synthetic methodologies, which are essential for discovering new drug candidates and agrochemicals. In this study, we present a unified strategy for a one-pot, catalytic enantioselective synthesis of α-alkyl and α,α′-dialkyl pyrrolidine, piperidine, and indolizidine alkaloids using readily available amides and alkynes. This synthesis is enabled by the identification and development of an Ir/Cu/*N*-PINAP catalyzed highly enantioselective and chemoselective reductive alkynylation of α-unbranched aliphatic amides, which serves as the key reaction. This reaction is combined with Pd-catalyzed tandem reactions in a one-pot approach, enabling the collective, catalytic enantioselective total syntheses of eight alkaloids and an anticancer antipode with 90–98% *ee*. The methodology's enantio-divergence is exemplified by the one-step access to either enantiomer of alkaloid bgugaine.

Efficiency, selectivity, and generality are major goals in contemporary synthetic organic chemistry, particularly important for total synthesis of natural products and the development of new medicines, agrochemicals, and materials[1–3]. To achieve these goals, concepts such as step-economy[3], pot-economy[4], catalytic enantioselective total synthesis, unified strategy[5], and collective synthesis[1] have been advanced. Central to these strategies are key reactions or key steps[6], such as cycloadditions/annulations[1], one-pot sequential/tandem/cascade reactions[1], and multicomponent reactions[7]. These generally feature simultaneous formation of at least two bonds in target molecules. Although computer-aided key step identification has been introduced recently[8], it relies on known key reactions, highlighting the demand for novel key reactions/steps for highly efficient total synthesis of natural products, new medicines, and agrochemicals.

Pyrrolidine and piperidine alkaloids, as well as fused bicyclic frameworks like indolizidine alkaloids, are prevalent in both plant[9] and animal[10,11] kingdoms. Ant venom alkaloids[10] and over 800 alkaloids isolated from the skin of poison frogs as of 2005[11] mostly possess a stereogenic unbranched α-alkyl or a *cis*- or *trans*-α,α′-dialkyl pyrrolidine/piperidine motif (Fig. 1f). Such structural motifs are also found in medicinal agents[12]. Although preliminary biological studies on some alkaloids reveal diverse biological activities such as cytotoxic, insecticidal, hemolytic, antibacterial, antifungal, and necrotic properties[10,11,13], comprehensive investigation has been hampered by a lack of samples from natural sources. Consequently, developing methods for enantioselective synthesis of these alkaloids has been a significant objective in organic synthesis[14–20]. However, most reported methods rely on stoichiometric chiral pools and chiral auxiliaries[16], necessitating long synthetic sequences for transforming commercially available materials into target molecules[5]. The development of catalytic asymmetric reactions has led to elegant total syntheses of several pyrrolidine, piperidine, and indolizidine alkaloids[17–19].

Despite the importance of pyrrolidine, piperidine, and indolizidine alkaloids, only a few highly efficient total syntheses of these compounds have been reported. Between 2012 and 2013, Kroutil et al. documented a multi-enzymatic cascade process for the three-step

[1]Department of Chemistry and Fujian Provincial Key Laboratory of Chemical Biology, College of Chemistry and Chemical Engineering, Xiamen University, Xiamen, Fujian 361005, P. R. China. ✉e-mail: pqhuang@xmu.edu.cn

**Fig. 1 | Background and summary of this work. a** Reported racemic direct reductive alkylation of amides. **b** The enantioselective reductive alkynylation of tertiary benzamides: method and limitation. **c** The enantioselective reductive alkynylation/alkylation of secondary amides: method and limitation. **d** Challenging substrates and targets for the highly efficient catalytic enantioselective total synthesis of alkaloids based on the catalytic enantioselective reductive alkylation/alkynylation of amides. **e** This work: The key reaction and its application to the general, one-pot, catalytic enantioselective syntheses of pyrrolidine piperidine, and indolizidine alkaloids. **f** Representative alkaloids/achieved targets: eight alkaloids and one anticancer antipode.

total synthesis of either enantiomer of isosolenopsin (**A-3**) from commercially available compounds[20,21]. In 2013, Glorius' group developed a four-step total synthesis of *ent*-monomorine I (*ent*-**A-8**) based on ruthenium-NHC-catalyzed asymmetric hydrogenation of indolizines[22]. However, due to the inherent substrate specificity of enzymatic reactions, the former method was limited to the synthesis of 2-methylpiperidine/pyrrolidine alkaloids, and the latter allowed access only to the unnatural enantiomer of monomorine I. Consequently, achieving highly efficient and general total synthesis of unbranched α-alkyl and α,α′-dialkyl pyrrolidine, piperidine, and indolizidine alkaloids with high enantioselectivity (≥90% *ee*) remains a significant challenge, despite recent advancements in the catalytic asymmetric synthesis of chiral α-alkylamines and allyl/propargylamines[23–25].

Amides are a class of bench-stable, readily available compounds widely used for synthesizing substituted amine motifs in the context of total synthesis of alkaloids and medicinal agents[14,15]. Nonetheless, due to the high stability of amides, their transformation into substituted amines requires multiple synthetic steps[5]. Over the past decade, the direct reductive functionalization of amides to yield amines (Fig. 1a) has emerged as a powerful strategy for total synthesis of alkaloids[26–38]. Despite the significant progress, the direct catalytic reductive alkylation of amides to yield chiral amines remains a formidable challenge. Recently, Huang and Wang established an Ir/Cu tandem catalysis protocol[39] and an Ir/Cu/organocatalyst multicatalysis protocol[40] for the enantioselective reductive alkynylation of amides (Fig. 1b, c). However, the former (Fig. 1b) is restricted to the transformation of tertiary benzamide derivatives[39], and the latter (Fig. 1c), while broader in scope, does not permit the highly enantioselective reductive alkynylation of unbranched aliphatic secondary amides[40]. Therefore, the catalytic asymmetric reductive alkynylation of aliphatic tertiary amides, particularly the α-unbranched ones with high enantioselectivity, remains an unresolved challenge (Fig. 1d).

## Results

### Synthetic design
To develop a highly efficient and universal enantioselective approach to unbranched α-alkyl and α,α′-dialkyl pyrrolidine, piperidine, and indolizidine alkaloids, it was necessary to identify and establish a key reaction. To accomplish this, several challenges needed to be addressed: (1) The use of protecting groups should be minimized, and multi-step functional group manipulations should be avoided. Ideally, several reactions, including deprotection, could be performed sequentially in a single pot; (2) A versatile approach is required to access diverse targets with significant structural variation, including mono- and bicyclic, five- and six-membered saturated nitrogen heterocycles with varying substituents and the presence or absence of an *N*-methyl group; (3) The current state of organic synthesis does not allow for direct, catalytic enantioselective reductive alkylation of carbonyl compounds with Grignard or organolithium reagents; (4) As previously mentioned, the catalytic asymmetric reductive alkynylation of aliphatic tertiary amides, particularly the α-unbranched ones, has yet to achieve high enantioselectivity (Fig. 1d).

We hypothesized that by leveraging the rich chemistry of alkynes[41–43], we could develop a versatile and highly enantioselective method for the direct reductive alkynylation of unbranched aliphatic amides. This would create a general and highly efficient entry point to the diverse alkaloids depicted in Fig. 1f. In this context, the formation of the C−C bond through catalytic asymmetric reductive alkynylation would not only establish the first asymmetric center but also induce the formation of other stereogenic centers during C−N bond formation (intramolecular reductive amination), constituting a key reaction. Moreover, this methodology highlights the use of alkynes as surrogates for alkyl carbanions[44], ensuring the necessary chemoselectivity and generality.

We now report a general amide/alkyne-based reductive alkynylation/annulation strategy (referred to as AARA methodology) for the one-pot asymmetric synthesis of diverse α-alkyl or α,α′-dialkyl pyrrolidine, piperidine, and indolizidine alkaloids from aliphatic tertiary amides and terminal alkynes (Fig. 1e).

### Catalytic enantioselective reductive alkynylation of tertiary aliphatic amides: reaction development
Upon identifying the crucial reaction, our strategic focus revolved around developing a general and highly enantioselective catalytic reductive alkynylation process for α-unbranched aliphatic amides. We chose *N,N*-dibenzylisobutyramide (**1a**) as our test aliphatic amide, and phenylacetylene (**2a**) as the alkyne partner, to investigate the catalytic asymmetric reductive alkynylation. Initial screening involved chiral ligands. Various chiral ligand classes have been employed in catalytic asymmetric alkynylation reactions to yield propargylamines[25,45–48], including Li's tridentate ligand pyridine bisoxazoline [PyBox, (*S,S*)-**L2**][45], Knochel's chiral bisphosphine ligand [(*R,R*)-Quinoxp, **L5**][46], Carreira's *N*-PINAP (**L4**)[47], and Ma's recently developed Pyrinap ligands[48]. We selected some cost-effective, commercially available ligands and tested a few new ones.

We initially established an efficient protocol consisting of sequential treatment of amide **1a** with Vaska's complex [IrCl(CO)(PPh₃)₂] (1.0 mol%), tetramethyldisiloxane[49] (TMDS, 2.0 equiv) at room temperature for 10 min in toluene, followed by phenylacetylene (**2a**, 1.2 equiv), CuBr (5.0 mol%), and a chiral ligand (5.0 mol%). After evaluating several chiral ligand classes (Table 1), including those based on (*S*)-(+)-BNPA, **L1**, PyBox [(*S,S*)-**L2**], bisoxazoline [BOX, (*S,S*)-**L3**], (*R,P*)-*N*-PINAP (**L4**), (*R,R*)-QuinoxP* (**L5**), and Zhang's chiral Ming-Phos **L6**[50], (*R,P*)-*N*-PINAP [(*R,P*)-**L4**]] emerged as the most promising, delivering propargylic amine **3a** with an excellent enantiomeric excess of 93% but a moderate 58% yield (Table 1, entry 1).

Encouraged by this outcome, we assessed various conditions for reaction optimization, with key results summarized in Table 1. Solvent choice influenced the asymmetric reaction (see Table 1, entries 1–5), and the mixed solvent system of toluene/THF = 3:1 (*v/v*) proved optimal in terms of both yield (79%) and enantioselectivity (90% *ee*) (Table 1, entry 5). Multiple Cu(I) and Cu(II) copper salts demonstrated effectiveness (see Table 1, entries 5–10), with CuBr and CuI yielding comparable results (79%/80% yield; 90%/89% *ee*) (Table 1, entries 5 and 6).

We then investigated the impact of additives (see Table 1, entries 11–16). Employing 5.0 mol% triethylamine as an additive, in conjunction with CuBr or CuI, proved beneficial, yielding **3a** with 93% *ee* (91% yield, entry 11) and 92% *ee* (94% yield, entry 15), respectively. Interestingly, we discovered that for CuBr- and CuI-catalyzed reactions, *ee* could be further enhanced to 96% and 98%, respectively, by inversely adding the partial reduction mixture to a combination of NEt₃ and chiral ligand (*R,P*)-**L4** at 0 °C (Table 1, entries 17 and 18). Consequently, the optimized conditions for reductive alkynylation of amide **1a** were defined in entry 18 of Table 1. The absolute configuration of propargylamine **3a** was determined to be *S* by comparing optical rotation data with those reported[46,47].

### Scope of amides and alkynes
Due to the unexplored potential of catalytic, enantioselective reductive alkynylation in α-unbranched aliphatic amides and the necessity of resulting propargylamines as intermediates for alkaloid synthesis (Fig. 1), the reaction applicability was examined. As demonstrated in Fig. 2A, various α-unbranched *N,N*-dibenzylamides (**1b–1g**), including propionamide **1g**, reacted smoothly, yielding the corresponding propargylamines **3b–3g** in 79–94% yields and 90–95% enantiomeric excess (*ee*). It is important to note that a one-pot alkaloid synthesis strategy requires the use of amides with diverse *N,N*-substituents.

**Table 1 | Optimization of reaction conditions**

| Entry | Ligand | Solvent | Cu catal. | Additive | Yield (%)[a] | ee %[b] |
|---|---|---|---|---|---|---|
| 1 | L4 | Toluene | CuBr | – | 58 | 93 |
| 2 | L4 | THF | CuBr | – | 62 | 87 |
| 3 | L4 | CH$_2$Cl$_2$ | CuBr | – | 21 | 84 |
| 4 | L4 | Toluene:THF = 1:1 | CuBr | – | 74 | 88 |
| 5 | L4 | Toluene:THF = 3:1 | CuBr | – | 79 | 90 |
| 6 | L4 | Toluene:THF = 3:1 | CuI | – | 80 | 89 |
| 7 | L4 | Toluene:THF = 3:1 | CuCl | – | 46 | 79 |
| 8 | L4 | Toluene:THF = 3:1 | CuBr$_2$ | – | 64 | 82 |
| 9 | L4 | Toluene:THF = 3:1 | CuCl$_2$ | – | 71 | 76 |
| 10 | L4 | Toluene:THF = 3:1 | Cu(OAc)$_2$ | – | 43 | 81 |
| 11 | L4 | Toluene:THF = 3:1 | CuBr | Et$_3$N | 91 | 93 |
| 12 | L4 | Toluene:THF = 3:1 | CuBr | i-Pr$_2$NEt | 68 | 92 |
| 13 | L4 | Toluene:THF = 3:1 | CuBr | P(o-tolyl)$_3$ | 67 | 92 |
| 14 | L4 | Toluene:THF = 3:1 | CuBr | P(1-napthyl)$_3$ | 79 | 91 |
| 15 | L4 | Toluene:THF = 3:1 | CuI | Et$_3$N | 92 | 94 |
| 16 | L4 | Toluene:THF = 3:1 | CuI | P(1-napthyl)$_3$ | 85 | 93 |
| 17 | L4 | Toluene:THF = 3:1 | CuBr | Et$_3$N | 91 | 96[c] |
| 18 | L4 | Toluene:THF = 3:1 | CuI | Et$_3$N | 93 (91)[d] | 98[c] |

[a]Yield determined by GC.
[b]Determined by chiral HPLC analysis of the isolated product.
[c]Et$_3$N and/or L4 (5 mol%) were added to the mixture of 1a/Ir complex/TMDS at 0 °C.
[d]Isolated yield.

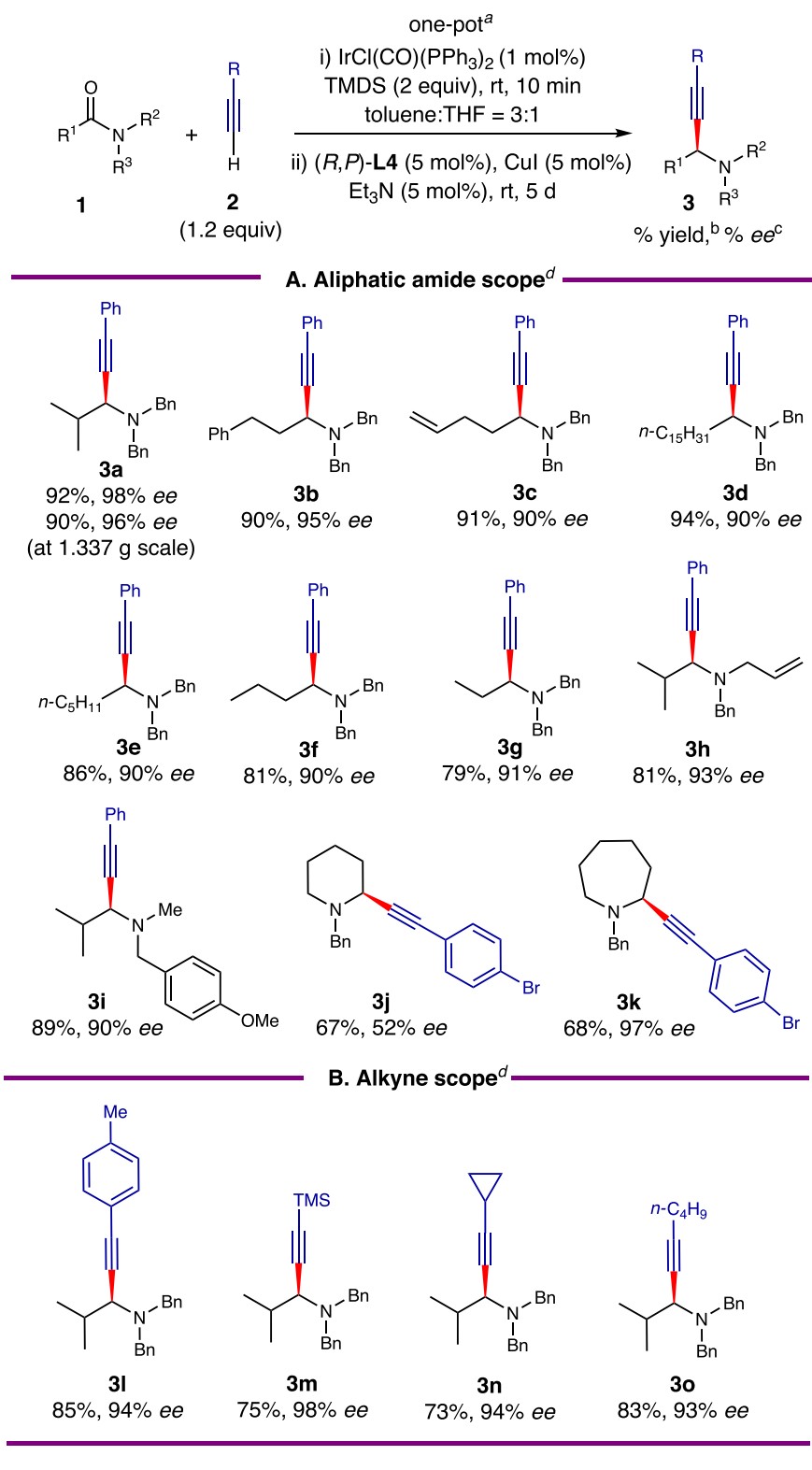

**Fig. 2 | Scope of Ir/Cu relay catalyzed enantioselective reductive alkynylation of aliphatic amides.** [a]Reaction conditions: amide **1** (0.4 mmol), IrCl(CO)(PPh$_3$)$_2$ (1 mol%), TMDS (0.8 mmol), toluene:THF = 3:1 (1 mL), rt, 10 min, 25 °C, then (*R,P*)-**L4** (5 mol%), CuI (5 mol%), Et$_3$N (5 mol%), alkyne **2** (0.48 mmol), toluene:THF = 3:1 (1 mL), rt, 5 d, 25 °C. b. Isolated yield. c. Determined by HPLC on a chiral stationary phase. [d]For the structures of amides and alkynes, see Tables 1 and 2 in the supplementary information. **A** Aliphatic amide scope. **B** Alkyne scope.

Consequently, *N*-allyl-*N*-benzylamide **1h** and *N*-Me-*N*-PMB amide **1i** were tested for asymmetric reductive alkynylation. To our delight, the resulting propargylamines **3h** and **3i** were produced with excellent *ee* (90% and 93%). The direct reductive alkynylation of cyclic amides was then investigated. Although the reductive alkynylation of six-membered lactam **1j** resulted in moderate *ee* (53%), cyclic propargylamine **3k** was formed with an outstanding *ee* (97%) when reacting with the seven-membered lactam **1k**.

The versatility of our strategy heavily relies on the flexibility of the alkyne partners. Therefore, we examined the scope of different alkynes (Fig. 2B). The reaction displayed insensitivity to acetylene substituents, as arylacetylene **2b**, trimethylsilylacetylene (**2d**), cyclopropylacetylene (**2e**), and alkylacetylene **2f** yielded analogous propargylamines **3l**–**3o** with 93–98% enantiomeric excess (*ee*).

## Chemoselectivity and functional group tolerance

Achieving chemoselectivity and functional group tolerance are two critical aspects for the success of our one-pot strategy. Consequently, we investigated the reactions of various functionalized amides containing either ester or keto groups, with the results summarized in Fig. 3, part A. The reaction accommodated ester (**1l**), cyclic ketone (**1m**), aryl ketone (**1n** and **1o**), and aliphatic (**1p**–**1r**) keto groups, yielding the corresponding functional propargylamines **3t**–**3y**. Furthermore, the alkyne partner was able to support different functional groups, such as a chlorine substituent in 4-chlorobut-1-yne (**2i**, Fig. 3, part B).

## One-pot, catalytic asymmetric total synthesis of eight pyrrolidine, piperidine, and indolizidine alkaloids and an anticancer unnatural enantiomer from amides and alkynes

After developing a versatile and highly chemoselective catalytic asymmetric reductive alkylation of amides, we were well-positioned to explore its application in the concise enantioselective synthesis of biologically active alkaloids. To begin, we conducted a gram-scale synthesis of propargylamine **3a**. A 5 mmol (1.337 g) scale reaction of **1a** yielded 1.589 g of **3a** (90% yield) with outstanding enantioselectivity (96% *ee*) (Table 1).

Next, our attention turned to the catalytic asymmetric total synthesis of alkaloids found in ant venom and poison-frog. As previously mentioned, ant venoms and poison-frog skins are rich sources of 2,6-dialkylpiperidines, 2,5-dialkylpyrrolidines, and indolizidine alkaloids. For instance, *cis*-2-methyl-6-nonyl-piperidine[51] (**A-3**), *cis*-2-methyl-6-undecyl-piperidine[52] (**A-4**), and the *N*-methyl derivative: *cis*-1,2-dimethyl-6-nonyl-piperidine[53] (**A-6**), as well as 2,5-dialkylpyrrolidines like 2-*n*-butyl-5-*n*-heptylpyrrolidine (**A-2**) are ant venom alkaloids isolated from various fire ant species[910].

Intriguingly, *cis*-2-methyl-6-nonyl-piperidine[11] (**A-3**), *cis*-2-methyl-6-undecyl-piperidine[11] (**A-4**), and 2-*n*-butyl-5-*n*-heptylpyrrolidine (**A-2**) have also been extracted from amphibian skin, where they are designated as 225I[11], 253J[11], and 225C[11], respectively. Among these alkaloids, the absolute configurations of those sourced from three ant samples—S. *geminutu* workers, S. *invictu* workers, and S. *invictu* alates—have been determined. However, the configurations of alkaloids isolated from other origins have not yet been ascertained due to sample scarcity, and they may differ based on possible variations in biosynthetic pathways[51].

We chose the alkaloids *cis*-2-methyl-6-nonyl-piperidine (*cis*-225I)[52] (**A-3**) and *cis*-2-methyl-6-undecyl-piperidine (*cis*-253J)[53] (**A-4**) as our initial targets, as they are potent inhibitors of neuronal nitric oxide synthase and of [3*H*]-perhydrohistrionicotoxin binding to sites associated with the nicotinic receptor-gated ion channel in Torpedo california electric organ, respectively. To determine the enantioselectivity of the one-pot total synthesis, we first explored a two-step synthesis (Fig. 4). Consequently, δ-keto amide **1p**, easily prepared in a single step from commercially available 5-oxo-hexanoic

acid, was subjected to chemoselective and enantioselective alkynylation with non-1-yne (**2j**), yielding propargylamine ketone **3z** in 90% yield and 98% *ee*. Scaling up the reaction to 1.238 g (4.0 mmol) provided **3z** in 86% yield and 92% *ee*. In the presence of concentrated HCl, Pd(OH)$_2$/C-catalyzed tandem hydrogenation-hydrogenolysis and reductive cyclization of keto-propargylamine **3z** yielded the desired (−)-*cis*-2-methyl-6-nonyl-piperidine (*cis*-225I) (**A-3**), isolated as hydrochloride salt, in 90% yield and excellent *cis*-diastereoselectivity (*dr* > 30:1).

Encouraged by this outcome, we undertook a one-pot tandem reductive alkynylation-hydrogenation-*N*-debenzylation-reductive cyclization of δ-keto amide **1p**, resulting in alkaloid (−)-*cis*-2-methyl-6-nonyl-piperidine (*cis*-225I) (**A-3**) in one-pot with 71% yield and 30:1 *dr* in favor of the *cis*-diastereomer. The enantiomeric excess of (−)-**A-3** was deduced from the intermediate **3z** to be 92%.

In subsequent investigations, we first conducted the catalytic enantioselective reductive alkynylation to determine the *ee* of the propargylamine intermediate and subsequently undertook the one-pot tandem reaction to synthesize the natural product. Thus, the one-pot tandem reaction of keto amide **1p** with undec-1-yne (**2g**) produced alkaloid (−)-*cis*-2-methyl-6-undecyl-piperidine (253J) (**A-4**) in 72% yield and >20:1 *dr* (*cis/trans*). The *ee* of (−)-*cis*-**A-4** was 90%, deduced from that of the keto amine **3v**.

While numerous enantioselective syntheses of 2-methyl-6-alkyl-piperidine ant venom alkaloids like **A-3** and **A-4** have been reported[20,21,54], synthesis of poison-frog alkaloid *cis*-197F[11] (**A-5**), a 2-ethyl-6-alkyl-piperidine alkaloid, has not been reported. Extending our methodology to keto amide **1q** and hex-1-yne (**2f**) using (*R,P*)-**L4** as the chiral ligand directly afforded alkaloid (−)-*cis*-197F (**A-5**) in 62% yield with >20:1 *dr* (*cis/trans*) and 91% *ee*, determined on **3w**.

The one-pot tandem reaction could also be applied to the synthesis of 2,5-*cis*-dialkylpyrrolidine alkaloids. Unlike their piperidine counterparts, 2,5-*cis*-dialkylpyrrolidines are rare among ant venom alkaloids[10,55]. Interestingly, while 2-butyl-5-heptylpyrrolidine (**A-2**) from the poison gland of the South African ant *Solenopsis punctaticeps* was identified as the *trans*-diastereomer, the stereochemistry of the same alkaloid found in one Colombian population of *Dendrobates histrionicus*, named with the code 225C, has not yet been determined[11,56]. Since little enantioselective synthesis of *cis*-2-butyl-5-heptylpyrrolidine *cis*-225C (**A-2**) has been reported[57] and a racemic synthetic product of *cis*- and *trans*-diastereomers was found to be a high-affinity ligand[58], we decided to undertake a one-step synthesis of this alkaloid. Employing (*R,P*)-**L4** as the chiral ligand, the reductive coupling of γ-keto amide **1r** with hept-1-yne (**2h**) in tandem with reductive cyclization produced pyrrolidine alkaloid (−)-*cis*-225C (**A-2**) in 65% yield with >20:1 *dr* (*cis/trans*) and 90% *ee*, assessed at the propargylamine **3x** stage.

It is important to note that *N*-methyl piperidine and *N*-methyl pyrrolidine structures are found in certain alkaloids. Our methodology's adaptability to *N*-substituents provides a direct approach to these structural motifs, where the *N*-methyl group is incorporated as part of the starting material. Using (*R,P*)-**L4** as the chiral ligand, the one-pot reaction of *N*-benzyl-*N*-methyl-5-oxohexanamide **1s** with non-1-yne (**2j**) directly produced the *N*-methylpiperidine alkaloid (−)-*cis*-1,2-dimethyl-6-nonyl-piperidine (**A-6**) in 76% yield, with a >20:1 *dr* (*cis/trans*) and 94% *ee*, as determined on **3aa**.

Next, we focused on the enantioselective synthesis of (−)-bguaine (**A-1**) (Fig. 5), a plant alkaloid derived from *Arisarum vulgare* tubers, known for its antibacterial activity against Gram-positive bacteria and antimycotic activity against some *Candida* and *Cryptococcus* strains. Employing (*R,M*)-**L4** as the chiral ligand in the reductive alkynylation process, the one-pot reaction of simple amide **1t** with commercially available 3,3-diethoxyprop-1-yne (**2k**) yielded natural (−)-bguaine (**A-1**) with an 81% yield [94% *ee* for (−)-**3ab**] (Fig. 5). Since the unnatural enantiomer, (+)-bguaine (*ent*-**A-1**), has reported anticancer activity[59,60], its synthesis was also pursued[61]. Using chiral ligand

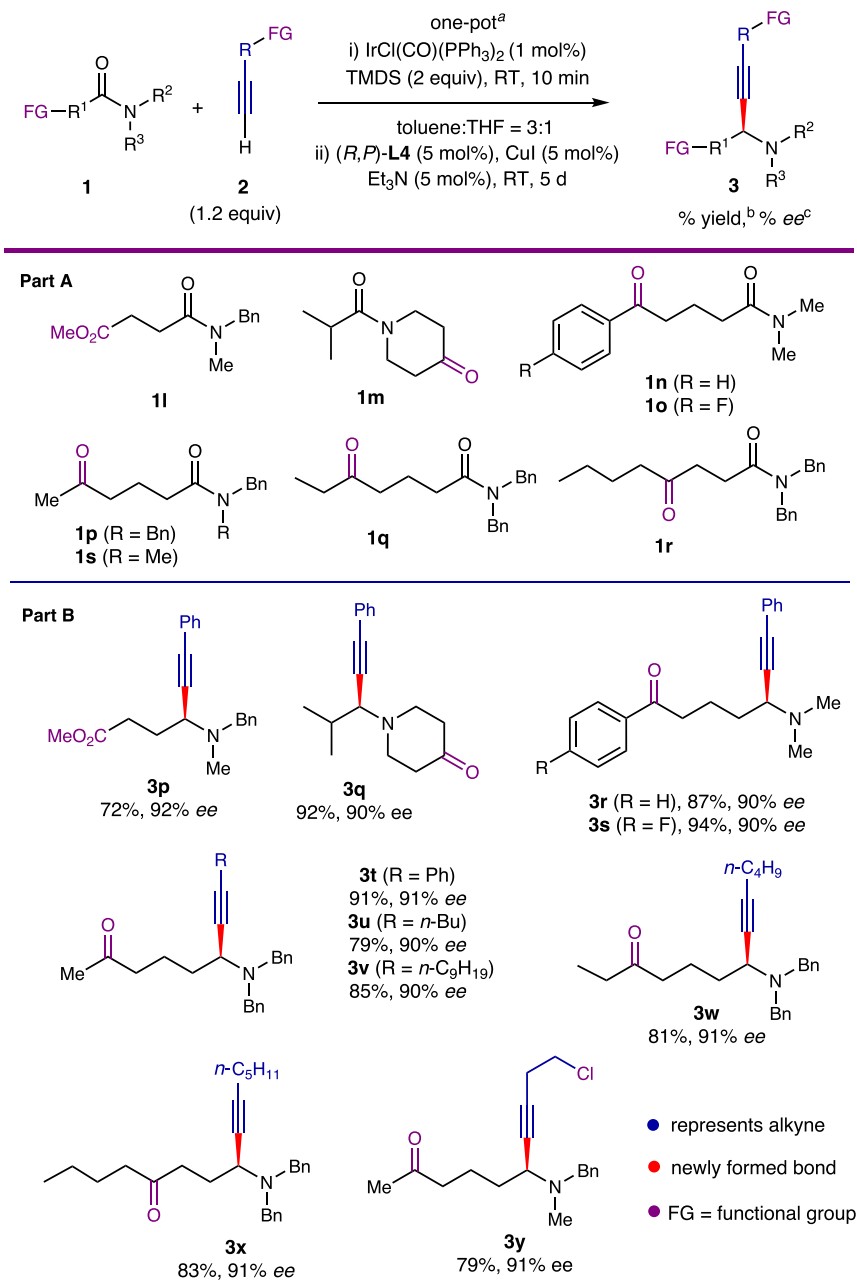

**Fig. 3 | Chemoselectivity and functional group tolerance of the catalytic enantioselective reductive alkynylation of aliphatic amides. Part A** Chemoselectivity and functional group tolerance of the amides. **Part B** Chemoselectivity and functional group tolerance of the alkyne. [a]For reaction conditions, see footnote a in Fig. 2. [b]Isolated yield. [c]Determined by HPLC on a chiral stationary phase.

(R,P)-N-PINAP [(R,P)-**L4**], the one-pot reductive transformation yielded (+)-bgugaine (ent-**A-1**) with an 82% yield and 92% ee, as determined on propargylamine intermediate (ent-)-**3ab**.

Lastly, we directed our attention to the asymmetric total synthesis of indolizidine alkaloids. To create the enantiomer represented by (−)-5,9Z-indolizidine alkaloid 209 D[18,62] (**A-7**), (R,M)-**L4** was employed as the chiral ligand for the reductive alkynylation. A combination of δ-keto amide **1u** with functionalized alkyne **2k** directly yielded indolizidine alkaloid (−)-5,9Z-indolizidine 209 D (**A-7**) with a 60% yield (90% ee, determined at the intermediate **3ac** stage). The spectral and physical data of our synthetic product matched the reported values.

In our final demonstration of the power of our methodology, the enantioselective total synthesis of (+)-monomorine I (**A-8**) was envisioned. This 5Z,9Z-3-methyl-5-butyl indolizidine alkaloid is a pharaoh ant trail pheromone isolated from *Monomorium pharaonis* L[63]. and later from the ant *Myrmicaria melanogaster* from Brunei[55]. In 1985, Husson and Royer achieved the first enantioselective total synthesis of (−)-monomorine I[64], determining the absolute configuration of the natural (+)-monomorine I as 3R,5S,9S. This alkaloid has since become the target of extensive synthetic efforts[19,65]. For the synthesis of (+)-monomorine I (**A-8**), (R,P)-**L4** was used as the chiral ligand. The one-pot asymmetric reaction of δ-keto amide **1p** with functionalized alkyne **2l** directly yielded alkaloid (+)-monomorine I (**A-8**) with a 51% yield, excellent ee (93%), and diastereoselectivity, predominantly forming the naturally occurring (5Z,9Z)-diastereomer. The spectral and physical data of our synthetic product matched the reported values[65]. Remarkably, this catalytic enantioselective tandem reaction formed one C–C bond and two C–N bonds, establishing the correct relative

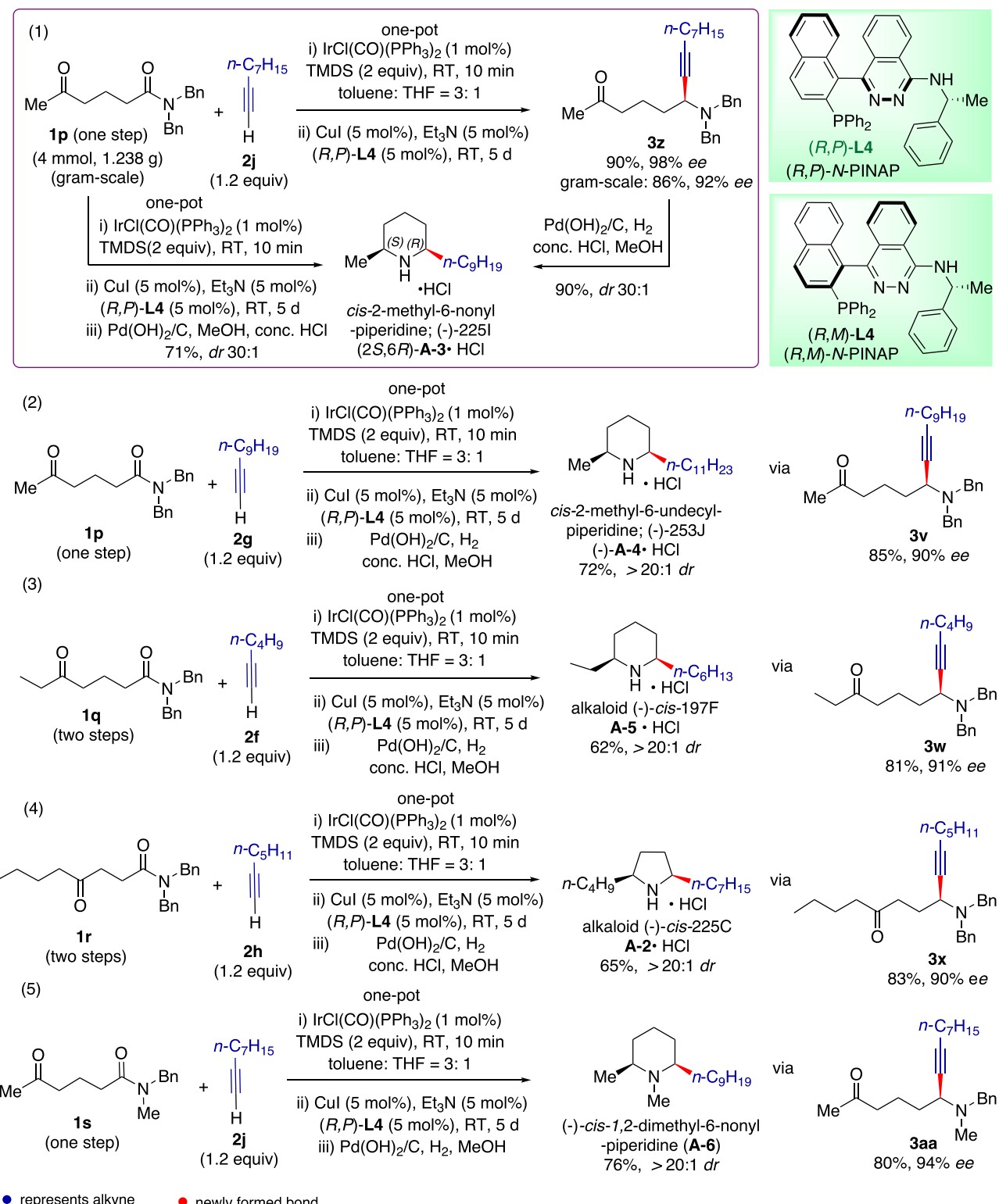

**Fig. 4 | One-pot, catalytic enantioselective total syntheses of pyrrolidine and piperidine alkaloids from δ/γ-keto amides and alkynes.** (1) One-pot synthesis of *cis*-225I (**A-3**). (2) One-pot synthesis of *cis*-253J (**A-4**). (3) One-pot synthesis of *cis*-197F (**A-5**). (4) One-pot synthesis of *cis*-225C (**A-2**). (5) One-pot synthesis of *cis*-1,2-dimethyl-6-nonyl-piperidine (**A-6**).

and absolute stereochemistries of three stereogenic centers, and producing one isomer from eight possible stereoisomers.

## Discussion
A plausible mechanism for the one-pot, catalytic enantioselective total synthesis of indolizidine alkaloid monomorine I is outlined in Fig. 6. The

first stereogenic center was established by the catalytic enantioselective reductive alkynylation of δ-keto amide **1p**. Under acidic and Pd-catalyzed hydrogenation/hydrogenolytic conditions, the cleavage of ketal protecting group, hydrogenation of alkynyl moiety, cleavages of two N-benzyl groups, and intramolecular reductive amination occurred sequentially to give intermediate **4a**, which could be isolated, and its

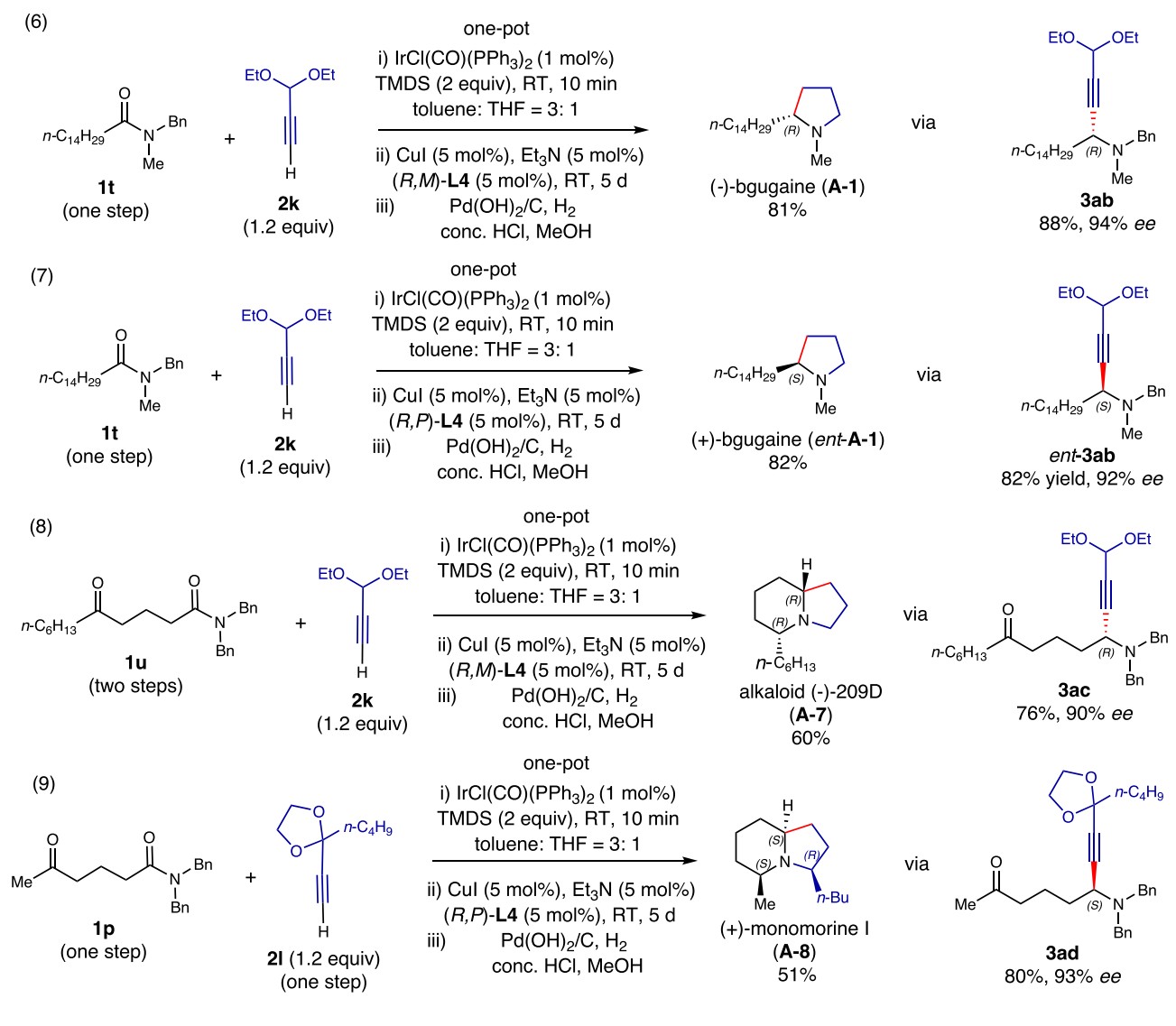

**Fig. 5 | One-pot, catalytic enantioselective total syntheses of pyrrolidine and indolizidine alkaloids from amides/δ-keto amides and functionalized alkynes.** (6) One-pot synthesis of (−)-bgugaine (**A-1**). (7) One-pot synthesis of (+)-bgugaine (*ent*-**A-1**). (8) One-pot synthesis of (−)-5,9Z-indolizidine 209 D (**A-7**). (8) One-pot synthesis of (+)-monomorine I (**A-8**).

structure was determined by comparison with the data reported for a racemic **4a** (see the Supplementary Information)[65]. Further one-pot intramolecular reductive amination occurred via presumed intermediate **4b** to deliver the final product **A-8**. The stereochemical course of the two intramolecular reductive aminations could be understood in terms of stereoelectronic effect as showcased by Stevens and Lee in their seminal racemic total synthesis of monomorine I[66].

In summary, we have established and refined a multifaceted approach for the enantioselective catalytic reductive alkynylation of linear aliphatic tertiary amides. Utilizing our highly effective Amide/Alkyne-based Reductive Annulation Strategy (AARA methodology), we have achieved a one-pot, catalytic enantioselective total synthesis of eight pyrrolidine, piperidine, and indolizidine alkaloids, as well as an anticancer antipode, with 90–98% enantiomeric excess (*ee*). Given that the amides/alkynes are either commercially available or can be synthesized in just one or two steps from readily accessible compounds, our approach represents the most concise and adaptable catalytic asymmetric total syntheses of the target alkaloids to date. This robust and highly stereoselective method sets the stage for the efficient

catalytic enantioselective total synthesis of other bioactive pyrrolidine, piperidine, indolizidine, and pyrrolizidine alkaloids, as well as therapeutically significant agents.

## Methods

### General procedure for the one-pot catalytic asymmetric reductive alkynylation of tertiary amides

To a flame-dried Schlenk tube were added CuI (3.8 mg, 0.02 mmol), (*R,P*)-*N*-PINAP [(*R,P*)-**L4**] (11.2 mg, 0.02 mmol) and toluene: THF = 3:1 (1 mL) under a N₂ atmosphere. After being stirred at room temperature for 5 min, triethylamine (6 μL, 0.02 mmol) and alkyne **2** (0.48 mmol) were added, and the resulting mixture was stirred at room temperature for 30 min.

To another flame-dried Schlenk tube were added sequentially IrCl(CO)(PPh₃)₂ (3.12 mg, 1 mol%), an amide **1** (0.4 mmol, 1 equiv), TMDS (144 μL, 0.8 mmol, 2 equiv) and toluene:THF = 3:1 (1 mL) under N₂ atmosphere at room temperature. After being stirred for 10 min, the resulting mixture was added to the abovementioned Schlenk tube containing CuI, (*R,P*)-**L4**, triethylamine and a terminal alkyne at 0 °C.

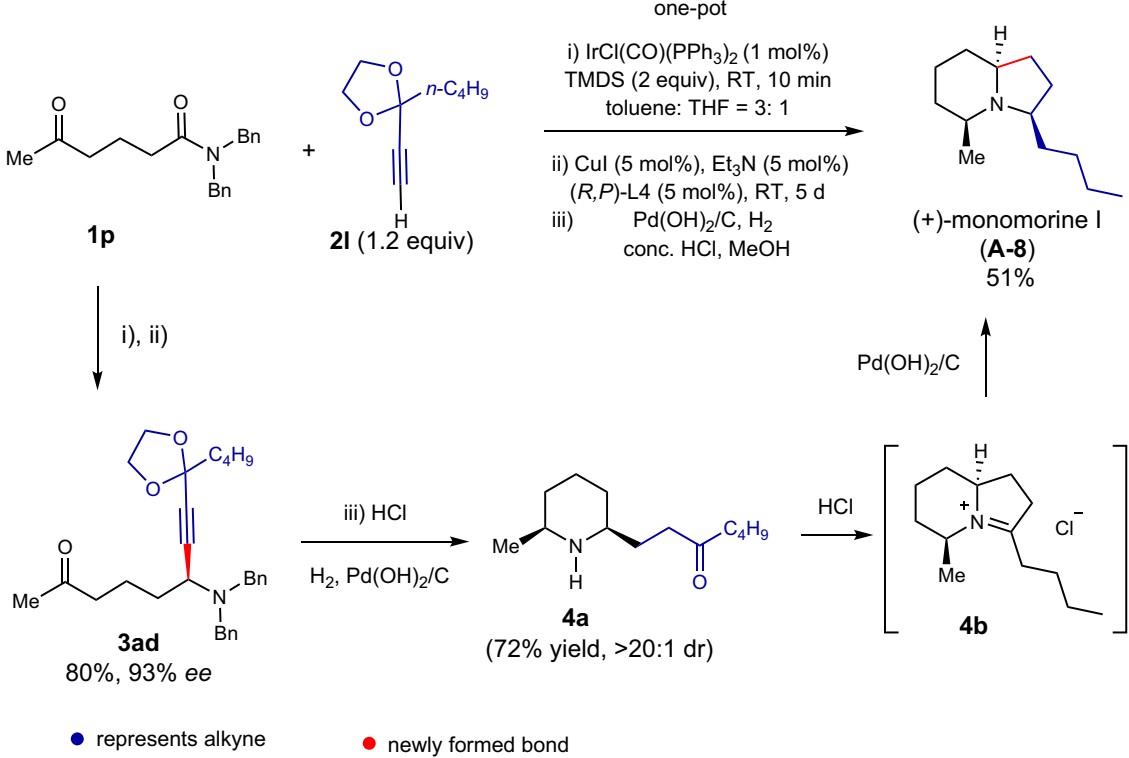

**Fig. 6 | Plausible mechanism.** Plausible mechanism for the one-pot, catalytic enantioselective total synthesis of indolizidine alkaloid monomorine I.

The mixture was stirred at room temperature for 5 d. The reaction mixture was concentrated under reduced pressure, and the residue was purified by flash chromatography on silica gel eluting with petroleum ether/ethyl acetate to afford the corresponding chiral propargylic amine **3**.

### General procedure for the one-pot catalytic asymmetric synthesis of alkaloids
To a flame-dried Schlenk tube were added CuI (9.5 mg, 0.05 mmol), (*R,P*)-**L4** (28.0 mg, 0.05 mmol), and toluene: THF = 3:1 (2 mL) under a $N_2$ atmosphere. The resulting mixture was stirred at room temperature for 5 min. Triethylamine (13 µL, 0.05 mmol) and a terminal alkyne **2** (1.2 mmol) were added, and the mixtures was stirred at room temperature for 30 min.

To another flame-dried Schlenk tube were sequentially added IrCl(CO)(PPh$_3$)$_2$ (7.8 mg, 1 mol%), an keto amide **1** (1.0 mmol, 1 equiv), TMDS (0.36 mL, 2 mmol) and toluene:THF = 3:1 (3 mL) under a $N_2$ atmosphere at room temperature. After being stirred for 10 min, the resulting mixture was added to the abovementioned Schlenk tube containing CuI, (*R,P*)-**L4**, triethylamine and alkyne at 0 °C. The mixture was stirred at room temperature for 5 d, then filtered through a short pad of Celite. The filtrate was concentrated under reduced pressure, and the residue was dissolved in MeOH (5 mL). To the resulting mixture, Pd(OH)$_2$/C was added, and the flask was purged three times with hydrogen. The suspension was stirred for 12 h at room temperature under a hydrogen atmosphere (1 atm). The resulting mixture was filtered through a short pad of Celite, and washed with methanol (50 mL). The filtrate was concentrated and the residue was purified by flash chromatography on silica gel to afford the corresponding alkaloid.

### Data availability
All data that support the findings of this study are available in the online version of this paper in the accompanying Supplementary information (including experimental procedures, compound characterization data, and spectra). All other data are available from the corresponding author upon request.

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

## Acknowledgements

We are grateful to the National Natural Science Foundation of China (21931010) and the National Key R&D Program of China (2017YFA0207302) for financial support. We thank Miss. Ting-Ting Chen for the preparation of some starting materials, and for Ms. Yan-Jiao Gao for technical assistance.

## Author contributions

P.-Q.H. conceived and directed the project and wrote the paper. F.-F.X., J.-Q.C., and D.-Y.S. performed the experiments and analyzed the data. All authors discussed the results and commented on the manuscript.

## Competing interests

The authors declare no competing interests.
