## [Peer Review File · Nature Communications]

REVIEWERS' COMMENTS

Reviewer #1 (Remarks to the Author):

Huang and his coworkers submitted the outcomes of the study “Catalytic Enantioselective Reductive Alkynylation of Amides Enables Unified One-Pot Synthesis of Pyrrolidine Piperidine and Indolizidine Alkaloids” to nature communication for publication. In this study, the enantioselective addition of Cu-acetylides in-situ generated from various alkynes, in the presence of a chiral N-PINAP ligand, to receptors formed from the in-situ treatment of tertiary alkyl amides with tetramethyl disiloxane (TDMS) catalyzed by Vaska’s Ir-complex. The highly enantio-enriched products received from this cascade reaction harbor a chiral propargyl amine with an aliphatic carbonyl group, enabling the subsequent transformations for the rapid synthesis of alkaloids.

Notwithstanding the generally good results, closely related synthetic methods that take advantage of the in-situ formation of imine and/or iminium receptors have been reported from the same research group alone with numerous typos and errors within the manuscript’s content, giving rise to a decline to accept this manuscript.

Reviewer #2 (Remarks to the Author):

Dr. Huang and co-workers reported a powerful, multifaceted approach for the enantioselective reductive alkynylation of linear aliphatic tertiary amide. Combined with the following transformation of the alkynyl group, enantioselective synthesis of eight pyrrolidine, piperidine, and indolizidine alkaloids was effectively achieved. Personally, I like this approach. This nice piece of work is built upon Dr. Huang's systematic and pioneering studies on enantioselective reductive functionalization of amide. In this work, a key challenge of enabling reductive alkynylation of alpha-unbranched aliphatic tertiary amide has been effectively resolved through catalyst development. This certainly represents a major advance in the area of enantioselective amide functionalization and will find broader application in areas of such of medicinal chemistry and drug design. Considering the general interests in amide functionalization of enantioselective synthesis of heterocycles, this concise, robust and broadly applicable method certainly deserves publication in a top journal such as Nat. Commun. Thus, publication of this work is highly recommended. The following minor issues should be addressed before publication.

1. page 6, line 148, change exceptional to excellent.

2. page 7, line 160, trimethylamine should read triethylamine.

3. page 8, table 1, the structure of L5 is incorrect.

4. page 14, line 266, use bold form for 1p.

Reviewer #3 (Remarks to the Author):

In this Communication, Huang and co-workers report a highly enantioselective Ir/Cu relay-catalyzed alkynylation of tertiary aliphatic amides and its elegant utilization in the concise preparation of a series of pyrrolidine, piperidine, and indolizidine alkaloids. One of the key findings is that Carrera's ligand N-PINAP shows excellent enantioselectivity compared to other ligands. The authors found that the reaction can be optimized in the mixed solvent system of toluene/THF. Such conditions can be applied to a wide range of substrate scopes, and the corresponding good to excellent results are obtained in Tables 2 and 3. As the catalytic asymmetric reductive alkynylation of aliphatic tertiary amide is challenging but highly rewarding in organic synthesis and medicinal chemistry, this work is very interesting and significant to organic chemistry readers. I highly recommend its acceptance after minor revisions.

Fig 1. e. For a better layout, Vaska's complex is suggested to be replaced with $\text{IrCl}(\text{CO})(\text{PPh}_3)_2$ as in b.

Fig 1. f. The current drawing of the alkyl side chain of bguanine needs to be clarified.

Page 11, line 216, chloro substituent (2i) also confuses readers. In Table 3, only 3y contains a chlorine substituent.

Page 14, line 266, ketoamide 1p, 1p should be bold.

Page 16, line 335, (R, M)-L4 should be (R, P)-L4.

Please explain the stereochemical outcome for the two new chiral centers of the alkaloid A-8 from 3ad in Fig 2.

廈門大學

XIAMEN UNIVERSITY

Department of Chemistry, Xiamen, Fujian 361005, P. R. China

Tel.(86)-(592)-2180992; Fax. (86)-(592) 2186400; E-mail: pqhuang@xmu.edu.cn

August 10, 2023

Dear reviewers,

Thank you very much for devoting much time and efforts on our manuscript entitled (revised): “Catalytic Enantioselective Reductive Alkynylation of Amides Enables One-Pot Syntheses of Pyrrolidine/ Piperidine and Indolizidine Alkaloids” (manuscript number: NCOMMS-23-19144-T). We are in debt for your valuable comments and questions, which are very helpful for us to improve our manuscript. The manuscript has been alternated in line with the requirements indicated by you. Our point-by-point response to your comments are as follows.

We hope that the revisions and our responses are satisfied, and the manuscript will be formally accepted for publication in Nature Communications. Thank you for your kind considerations.

With best regards,

Pei-Qiang HUANG, Ph D

Professor

College of Chemistry and Chemical Engineering

Xiamen University

XIAMEN, FUJIAN 361005

P. R. CHINA

Tel.: 86-592-2182240; Fax. 86-592-2189959;

E-mail: pqhuang@xmu.edu.cn

Group Homepage: <http://huangpeiqiang.xmu.edu.cn/en/Home.htm>

Reviewer #1

Huang and his coworkers submitted the outcomes of the study “Catalytic Enantioselective Reductive Alkynylation of Amides Enables Unified One-Pot Synthesis of Pyrrolidine Piperidine and Indolizidine Alkaloids” to nature communication for publication. In this study, the enantioselective addition of Cu-acetylides in-situ generated from various alkynes, in the presence of a chiral N-PINAP ligand, to receptors formed from the in-situ treatment of tertiary alkyl amides with tetramethyl disiloxane (TDMS) catalyzed by Vaska’s Ir-complex. The highly enantio-enriched products received from this cascade reaction harbor a chiral propargyl amine with an aliphatic carbonyl group, enabling the subsequent transformations for the rapid synthesis of alkaloids.

Notwithstanding the generally good results, closely related synthetic methods that take advantage of the in-situ formation of imine and/or iminium receptors have been reported from the same research group alone with numerous typos and errors within the manuscript’s content, giving rise to a decline to accept this manuscript.

Our response: This reviewer has overlooked the key issues of our previous and current outcomes. As we have indicated in the manuscript, (1) we have established the highly enantioselective catalytic reductive alkynylation of α -unbranched aliphatic amides, which can’t be achieved by any previous methods; (2) the reactions are enabled by a new catalytic system: Ir/Cu/N-PINAP; and (3) the reaction in turn enables the unified one-pot syntheses of simple and fused pyrrolidine and piperidine alkaloids from amides and alkynes, which sets a new standard in organic synthesis in terms of efficiency, enantioselectivity, and versatility, as indicated by the second reviewer.

Reviewer #2

Dr. Huang and co-workers reported a powerful, multifaceted approach for the enantioselective reductive alkynylation of linear aliphatic tertiary amide. Combined with the following transformation of the alkynyl group, enantioselective synthesis of eight pyrrolidine, piperidine, and indolizidine alkaloids was effectively achieved. Personally, I like this approach. This nice piece of work is built upon Dr. Huang's systematic and pioneering studies on enantioselective reductive functionalization of amide. In this work, a key challenge of enabling reductive alkynylation of alpha-unbranched aliphatic tertiary amide has been effectively resolved through catalyst development. This certainly represents a major advance in the area of enantioselective

amide functionalization and will find broader application in areas of such of medicinal chemistry and drug design. Considering the general interests in amide functionalization of enantioselective synthesis of heterocycles, this concise, robust and broadly applicable method certainly deserves publication in a top journal such as Nat. Commun. Thus, publication of this work is highly recommended. The following minor issues should be addressed before publication.

Our response: Many thanks to this respectable reviewer for getting insight into the outcomes presented in our manuscript, for her/his vision on the state-of-the-arts of organic synthesis, for her/his encourage, and for helping us to improve the manuscript by indicating the following issues.

1. page 6, line 148, change exceptional to excellent.

Our response: Changed as suggested throughout the manuscript (p. 6 and p. 14).

2. page 7, line 160, trimethylamine should read triethylamine.

Our response: Corrected with many thanks.

3. page 8, table 1, the structure of L5 is incorrect.

Our response: The structure of L5 in table 1 has been corrected. In addition, “(R,R)-Quinox” corrected to “(R,R)-QuinoxP* in both Table 1 and text”.

4. page 14, line 266, use bold form for 1p.

Our response: Corrected with many thanks.

Reviewer #3

In this Communication, Huang and co-workers report a highly enantioselective Ir/Cu relay-catalyzed alkynylation of tertiary aliphatic amides and its elegant utilization in the concise preparation of a series of pyrrolidine, piperidine, and indolizidine alkaloids. One of the key findings is that Carrera’s ligand N-PINAP shows excellent enantioselectivity compared to other ligands. The authors found that the reaction can be optimized in the mixed solvent system of to/THF. Such conditions can be applied to a wide range of substrate scopes, and the corresponding good to excellent results are obtained in Tables 2 and 3. As the catalytic asymmetric reductive alkynylation of aliphatic tertiary amide is challenging but highly rewarding in organic synthesis and medicinal chemistry, this work is very interesting and significant to organic chemistry readers. I highly recommend its acceptance after minor revisions.

Our response: Many thanks to this respectable reviewer for getting insight into the

outcomes presented in our manuscript, for her/his vision on the state-of-the-arts of organic chemistry, for her/his encourage, and for helping us to improve the manuscript by indicating the following issues.

Fig 1. e. For a better layout, Vaska's complex is suggested to be replaced with $\text{IrCl}(\text{CO})(\text{PPh}_3)_2$ as in b.

Our response: Replaced with many thanks.

Fig 1. f. The current drawing of the alkyl side chain of bbugaine needs to be clarified.

Our response: Clarified with many thanks.

Page 11, line 216, chloro substituent (**2i**) also confuses readers. In Table 3, only **3y** contains a chlorine substituent.

Our response: Thanks for noting this. All the structures of alkynes are listed in the SI. We now revise "such as a chloro substituent (**2i**)." to "such as a chlorine substituent in 4-chlorobut-1-yne (**2i**).".

Page 14, line 266, ketoamide **1p**, **1p** should be bold.

Our response: Corrected with many thanks.

Page 16, line 335, (R, M)-L4 should be (R, P)-L4.

Our response: Corrected to with many thanks.

Please explain the stereochemical outcome for the two new chiral centers of the alkaloid A-8 from **3ad** in Fig 2.

Our response: To get insight into the mechanism of the tandem reaction, in addition to **3ad**, we also isolated and characterized **4a** as an intermediate, which allowed us to suggest a plausible mechanism. The following discussion and Fig. 6 have been added to the main text under the new section Discussion, and the characterization and NMR spectra of **4a** (in addition to **3ad**) have been added to the SI.

A plausible mechanism for the one-pot, catalytic enantioselective total synthesis of indolizidine alkaloid monomorine I is outlined in Fig. 6. The first stereogenic center was established by the catalytic enantioselective reductive alkynylation of δ -keto amide **1p**. Under acidic and Pd-catalyzed hydrogenation/hydrogenolytic conditions, the

cleavage of ketal protecting group, hydrogenation of alkynyl moiety, cleavages of two N-benzyl groups, and intramolecular reductive amination occurred sequentially to give intermediate **4a**, which could be isolated, and its structure was determined by comparison with the data reported for a racemic **4a** (see the supplementary information)⁶⁵. Further one-pot intramolecular reductive amination occurred via presumed intermediate **4b** to deliver the final product **A-8**. The stereochemical course of the two intramolecular reductive aminations could be understood in terms of stereoelectronic effect as showcased by Stevens and Ley in their seminal racemic total synthesis of monomorphine I⁶⁶.

Fig. 6. Plausible mechanism for the one-pot, catalytic enantioselective total synthesis of indolizidine alkaloid monomorphine I.

(2*S*,6*S*)-**6a** was obtained as a pale yellow oil in 72% yield; [α]_D²⁵ - 17.3 (*c* 1, CHCl₃);

IR (film) $\tilde{\nu}$: 3350, 2956, 1461 cm⁻¹; ¹H NMR (400 MHz, Chloroform-*d*) δ 2.82-2.75 (m, 1H), 2.70-2.63 (m, 1H), 2.62 – 2.48 (m, 2H), 2.44-2.37(m, 2H), 1.98 – 1.75 (m, 3H), 1.74 – 1.63 (m, 2H), 1.58 – 1.50 (m, 2H), 1.35 – 1.20 (m, 9H), 0.90 (t, *J* = 7.3, 3H); ¹³C NMR (101 MHz, Chloroform-*d*) δ 210.8, 57.0, 53.3, 42.6, 38.9, 32.9, 30.5, 29.7, 25.9, 24.1, 22.4, 21.9, 13.9; HRMS (ESI) *m/z* for C₁₃H₂₆NO ([M+H]⁺):212.2009, Found:

212.2003;

The editor indicated that: “one reviewer expressed concern to the editor's office about the veracity of the assignments from HPLC chromatograms, as they mentioned not all

retention times were matched. At least one (*ent-3ab*) does not show a baseline. Some appear out of focus due to what appears to be screenshot limitations. Please double-check your HPLC spectra before resubmission and include English translations of the data descriptions if they are necessary to understand the figure.”

Our response: We thank this reviewer for arising this problem that is very helpful for us. We have double checked all HPLC chromatograms have been and those unclear have been updated with the original ones (without modification). The replaced chromatograms include: **3a** (SI, on page 12), **3i** (SI, on page 20), **3m** (SI, on page 24), **3z** (SI, on page 37), **3u** (SI, on page 32), **3aa** (SI, on page 38).

For compound *ent-3ab*, both the original diagram for racemic compound (up dated with that determined at the same period of time as for the enantiomeric form) and the diagram (with base line) of the enantiomeric-enriched compound **3ab** (SI, on page 39). For most cases, good agreement of the retention times for racemic compound and the enantiomeric-enriched compound are observed. In some cases, a difference of 0.6 min was observed, which is acceptable as compared with that of another paper [compound **4aha** in Ref. 48: *Nat. Commun.* **12**, 19 (2021)]. Such difference is attributable to both the low polarity of eluent used (n-hexane/ isopropanol = 99.5: 0.5, v/v) and stability of the HPLC apparatus.

Thus, we can conclude that our HPLC data are veracity and reliable.

HPLC traces of *ent-3ab*

数据文件: C:\CHEM32\1\DATA\XFF\DEF_LC-LCY 2022-04-01 00-01-46\XFF-3-113-AD-00C-254.D
 样品名称: XFF-3-113 RE AD-00c-0.5V-254

=====

操作者 : LGS 序列号 : 3
 仪器 : 仪器 1 位置 : 样品瓶 07
 进样日期 : 2022-4-1 0:24:32 进样次数 : 1
 进样量 : 5.000 µl

采集方法 : C:\CHEM32\1\DATA\XFF\DEF_LC-LCY 2022-04-01 00-01-46\00C-60MIN-0.5V-254NM.M
 最后修改 : 2022-3-31 23:50:22 : LGS
 分析方法 : C:\CHEM32\1\METHODS\WDP\10C-10MIN-254NM.M
 最后修改 : 2023-8-9 0:19:51 : WDP (调用后修改)

=====
 面积百分比报告
 =====

排序 : 信号
 乘积因子 : 1.0000
 稀释因子 : 1.0000
 内标使用乘积因子和稀释因子

信号 1: VWD1 A, Wavelength=254 nm

峰 #	保留时间 [min]	类型	峰宽 [min]	峰面积 [mAU*s]	峰高 [mAU]	峰面积 %
1	6.241	MM R	0.1950	1.70608	1.45844e-1	50.4579
2	6.927	MM R	0.2415	1.67512	1.15601e-1	49.5421

总量 : 3.38120 2.61445e-1

=====
 *** 报告结束 ***

仪器 1 2023-8-9 0:20:45 WDP 页 1/1

数据文件: C:\CHEM32\1\DATA\XFF\DEF_LC-LCY 2022-04-01 00-01-46\XFF-3-113 CHIRAL AD-00C-254.D
 样品名称: XFF-3-113 CHIRAL AD-01c-254

=====

操作者 : LGS 序列号 : 4
 仪器 : 仪器 1 位置 : 样品瓶 08
 进样日期 : 2022-4-1 1:25:21 进样次数 : 1
 进样量 : 5.000 µl

采集方法 : C:\CHEM32\1\DATA\XFF\DEF_LC-LCY 2022-04-01 00-01-46\00C-60MIN-0.5V-254NM.M
 最后修改 : 2022-3-31 23:50:22 : LGS
 分析方法 : C:\CHEM32\1\METHODS\WDP\10C-10MIN-254NM.M
 最后修改 : 2023-8-9 16:09:45 : WDP (调用后修改)

=====
 面积百分比报告
 =====

排序 : 信号
 乘积因子 : 1.0000
 稀释因子 : 1.0000
 内标使用乘积因子和稀释因子

信号 1: VWD1 A, Wavelength=254 nm

峰 #	保留时间 [min]	类型	峰宽 [min]	峰面积 [mAU*s]	峰高 [mAU]	峰面积 %
1	6.373	MM R	0.2256	6.99114	5.16386e-1	96.1666
2	7.543	MM R	0.1811	2.78680e-1	2.56433e-2	3.8334

总量 : 7.26982 5.42030e-1

=====
 *** 报告结束 ***

仪器 1 2023-8-9 16:09:47 WDP 页 1/1